# Preparation of High-Performance Transparent Al_2_O_3_ Dielectric Films via Self-Exothermic Reaction Based on Solution Method and Applications

**DOI:** 10.3390/mi15091140

**Published:** 2024-09-11

**Authors:** Xuecong Fang, Honglong Ning, Zihan Zhang, Rihui Yao, Yucheng Huang, Yonglin Yang, Weixin Cheng, Shaojie Jin, Dongxiang Luo, Junbiao Peng

**Affiliations:** 1Guangdong Basic Research Center of Excellence for Energy & Information Polymer Materials, State Key Laboratory of Luminescent Materials and Devices, School of Materials Sciences and Engineering, South China University of Technology, Guangzhou 510640, China; 202030270402@mail.scut.edu.cn (X.F.); ninghl@scut.edu.cn (H.N.); 201930160189@mail.scut.edu.cn (Z.Z.); yucheng_h@163.com (Y.H.); 202420119057@mail.scut.edu.cn (Y.Y.); ms_weixin_cheng@mail.scut.edu.cn (W.C.); 18770602788@163.com (S.J.); psjbpeng@scut.edu.cn (J.P.); 2Huangpu Hydrogen Innovation Center/Guangzhou Key Laboratory for Clean Energy and Materials, School of Chemistry and Chemical Engineering, Guangzhou University, Guangzhou 510006, China; 3Guangdong Provincial Key Laboratory of Optical Information Materials and Technology, South China Academy of Advanced Optoelectronics, South China Normal University, Guangzhou 510006, China; 4Key Laboratory of Optoelectronic Devices and Systems of Ministry of Education and Guangdong Province, College of Physics and Optoelectronic Engineering, Shenzhen University, Shenzhen 518060, China

**Keywords:** low temperature, self-exothermic reaction, MIM device, transparent, low leakage current, high dielectric constant

## Abstract

As the competition intensifies in enhancing the integration and performance of integrated circuits, in accordance with the famous Moore’s Law, higher performance and smaller size requirements are imposed on the dielectric layers in electronic devices. Compared to vacuum methods, the production cost of preparing dielectric layers via solution methods is lower, and the preparation cycle is shorter. This paper utilizes a low-temperature self-exothermic reaction based on the solution method to prepare high-performance Al_2_O_3_ dielectric thin films that are compatible with flexible substrates. In this paper, we first established two non-self-exothermic systems: one with pure aluminum nitrate and one with pure aluminum acetylacetonate. Additionally, we set up one self-exothermic system where aluminum nitrate and aluminum acetylacetonate were mixed in a 1:1 ratio. Tests revealed that the leakage current density and dielectric constant of the self-exothermic system devices were significantly optimized compared to the two non-self-exothermic system devices, indicating that the self-exothermic reaction can effectively improve the quality of the dielectric film. This paper further established two self-exothermic systems with aluminum nitrate and aluminum acetylacetonate mixed in 2:1 and 1:2 ratios, respectively, for comparison. The results indicate that as the proportion of aluminum nitrate increases, the overall dielectric performance of the devices improves. The best overall performance occurs when aluminum nitrate and aluminum acetylacetonate are mixed in a ratio of 2:1: The film surface is smooth without cracks; the surface roughness is 0.747 ± 0.045 nm; the visible light transmittance reaches up to 98%; on the basis of this film, MIM devices were fabricated, with tested leakage current density as low as 1.08 × 10^−8^ A/cm^2^ @1 MV and a relative dielectric constant as high as 8.61 ± 0.06, demonstrating excellent electrical performance.

## 1. Introduction

With the advanced development of integrated circuits, there is a growing demand for thinner dielectric layers in electronic components [1,2,3]. However, when the thickness of the amorphous silicon dielectric layer is reduced to below 3 nm, the gate leakage current caused by direct tunneling can reach up to 1 A/cm^2^ [4]. To reduce leakage current and increase the thickness of the dielectric layer, various high-k dielectric materials have been used, including Al_2_O_3_, HfO_2_, ZnO, and ZrO_2_ [5,6,7,8,9]. They also offer advantages such as high mobility, high transparency, and suitability for flexible substrates [10,11,12].

The preparation method of high-k gate dielectric layers is the classical vacuum method [13,14,15,16], resulting in excellent film uniformity and high quality. However, these equipment options are characterized by high costs and lengthy experimental durations. Therefore, in order to achieve low-cost, rapid, and high-quality preparation of dielectric layers, the solution method has attracted widespread attention and research [17,18,19,20]. However, in the process of film preparation using solution methods, in order to achieve the removal of organic ligands and densification of the film, the annealing temperature is typically higher than 400 °C or even higher [21,22]. This limits the preparation of dielectric films on flexible substrates.

Recently, many new low-temperature solution methods for preparing dielectric layers have been studied. Wangying Xu and her team prepared Al_2_O_3_ dielectric films using a ‘aqueous route’ with annealing at 250 °C [23]. The leakage current density is 2.9 × 10^−7^ A/cm^2^@1 MV/cm, the dielectric constant is 8.6, and the breakdown field strength is greater than 2.5 MV/cm. Jeong-Wan Jo employed a high-density UV (DUV) treatment-assisted exothermic process using low-pressure mercury lamps to prepare Al_2_O_3_/ZrO_2_ dielectric layers at 180 °C [24]. The leakage current density was reduced to 4.68×10^−9^ A/cm^2^@1 MV/cm. Sumei Wang used infrared irradiation for 40 min to induce the low-temperature decomposition of AlCl_3_ precursor solution into Al_2_O_3_ films, achieving a high dielectric constant [7,8] and a low leakage current (3.5 × 10^−8^ A/cm^2^@1 MV/cm) [25]. A breakdown field strength greater than 3 MV/cm was obtained. These low-temperature solution methods share a common characteristic, which is the utilization of photo-irradiation or other methods to assist the exothermic process based on exothermic reaction. A drawback is the need for constant adjustment of the power and wavelength of auxiliary equipment to achieve optimal settings.

In this paper, we utilize a low-temperature self-exothermic reaction based on the solution method to prepare high-performance Al_2_O_3_ dielectric thin films. In the exothermic reaction, metal nitrates act as strong oxidants, while acetylacetonates serve as strong reductants. The mixed sample undergoes a spontaneous and vigorous redox reaction upon reaching a relatively low annealing temperature, generating a substantial amount of heat to compensate for the energy required for the formation of M-O-M bonds, thereby producing high-performance dielectric films. This exothermic process requires only a constant-temperature heating device and no additional auxiliary equipment, and the low annealing temperature allows compatibility with flexible substrates and reduces energy consumption. This low-temperature solution method offers advantages such as low cost, adjustable composition, safety, and the ability to produce large-area films [26,27,28]. Here, this paper proposes a self-exothermic reaction, distinct from traditional exothermic reactions, where the process only requires the mixing of metal precursors containing fuel ligands and oxidant ligands without the need for additional fuel; this simple precursor solution composition is easy to control and replicate, facilitating the preparation of stable dielectric layers and MIM (Metal–Insulator–Metal) devices.

## 2. Experiment

### 2.1. Synthesis of Precursor Solution

All reagents, including aluminum nitrate hydrate (Al(NO_3_)_3_·9H_2_O, AlNO), aluminum triacetylacetone (C_15_H_21_O_6_Al, AlAC) and N,N-Dimethylformamide (C_3_H_7_NO, DMF), were purchased from Merck and required no further purification.

The preparation of the self-exothermic precursor solution is as follows: The total concentration is controlled at 0.2 M, which can be calculated as follows:c (AlNO) + c (AlAC) = 0.2 M(1)

Five samples of precursor solutions were prepared by mixing AlNO and AlAC in different ratios, as shown in Table 1. The solvent chosen is DMF. It was stirred magnetically for 24 h, followed by aging for 4 h at rest. Subsequently, the precursor solution was filtered using a 0.22 μm organic-phase syringe filter to obtain the solution for spin-coating.

In this paper, the five samples of precursor systems will be simplified, with their names shown in Table 1.

### 2.2. Dielectric Film Fabrication

ITO glass substrates were subjected to ultrasonic cleaning in a sequence of deionized water/isopropanol/deionized water/isopropanol, then dried at 80 °C for 12 h. Prior to spin-coating, the substrates were UV-treated for 20 min. Subsequently, 40 μL of the precursor solution was spin-coated on the substrates at 500 r/min for 5 s and 5000 r/min for 30 s. Finally, dielectric films were pre-annealed at 80 °C for 10 min and post-annealed at 185 °C for 1 h.

The weight and heat changes of the precursor solution were measured using a Thermogravimetric Analyzer (TG, DZ-TGA101, Nanjing, China) and Differential Scanning Calorimetry (DSC, DZ-DSC300C, Nanjing, China). The surface tension of the precursor solution was measured using Attension Theta Lite (TL200, BiolinScientific, Gothenburg, Sweden). The surface morphology of the dielectric films was observed using Laser Scanning Confocal Microscopy (LSCM, OLS50-CB, Tokyo, Japan) and Atomic Force Microscopy (AFM, BY3000, Nano Instruments, Guangzhou, China). The thickness, density, and roughness of the films were fitted using an X-ray Reflectometer (XRR, PANalytical Empyrean, Almelo, The Netherlands). The optical properties were studied using an Ultraviolet-Visible Spectrophotometer (UV-Vis, UV-3600 Shimadzu, Kyoto, Japan). The phase structure of the films was analyzed using an X-ray Diffractometer (XRD, PANalytical Empyrean, Almelo, The Netherlands), and the functional groups were characterized using Fourier Transform Infrared Spectroscopy (FTIR, ATR Accessory, Nexus, Madison, WI, USA).

### 2.3. MIM Device Fabrication

The electrical properties of dielectric films are typically characterized by preparing MIM devices [29]. MIM devices were fabricated with Al_2_O_3_ as the dielectric layer, as shown in Figure 1. Al electrodes were thermally deposited on the surface of the dielectric film, with an electrode area of 1.256 × 10^−3^ cm^2^. The current–voltage (I–V), capacitance–voltage (*C*-V), and capacitance–frequency (*C*-F) characteristics of MIM devices were measured using the semiconductor parameter analyzer (Primarius FS-Pro, Shanghai, China). In order to ensure the authenticity and universality of the experimental data, three test samples were prepared for each component, and two different test points were characterized for each sample. If the test results obtained are close, it indicates good uniformity of the film.

## 3. Result and Discussion

### 3.1. Surface Tension Tests Results

The surface tension tests of the five precursor solutions are shown in Figure 2. The results show that all five samples of precursor solutions have relatively average surface tension. The surface tensions of samples S1, S2, S3, S4 and S5 studied in this paper are 35.77 mN/m, 36.58 mN/m, 35.37 mN/m, 35.67 mN/m, and 36.76 mN/m, aligning with spin-coating thin films.

### 3.2. TG-DSC Test Analysis

The TG-DSC test results are shown in Figure 3. From the results, it is evident that three groups of self-exothermic precursor samples exhibit a distinctly different thermal behavior compared to the other two non-self-exothermic precursor samples. S3, S4, and S5 exhibit intense exothermic behavior near 160~180 °C. In contrast, S1 and S2 show no significant exothermic activity throughout the entire temperature range (30~500 °C).

The decomposition temperature of Al(NO_3_)_3_ is around 200 °C, resulting in the formation of Al_2_O_3_ and the release of a large amount of gasses such as NO, NO_2_, and O_2_ [30]. From Figure 3a, it can be observed that S1 experiences a sharp decrease in mass around 220 °C, corresponding to the production of a large amount of gas during the decomposition reaction. Therefore, the exothermic behavior of S1 around 221 °C is likely due to the decomposition reaction. S2 exhibits a weak endothermic peak around 200 °C, which is due to the melting and sublimation of AlAC at this temperature, with a significant mass loss resulting from the evaporation of a substantial amount of material.

However, three samples of different self-exothermic precursors exhibited a significant exothermic peak at 160~180 °C. S3, S4 and S5, respectively, exhibited sharp exothermic peaks around 160 °C, 170 °C, and 182 °C, with exothermic intensities of 4.46 mW/mg, 4.48 mW/mg and 6.53 mW/mg, respectively. Each exothermic peak corresponds to a significant mass loss, which is due to the generation of large amounts of gas from redox reactions. The temperature corresponding to the exothermic peak increases with an increase in the proportion of AlAC; at the same time, the exothermic intensity also increases. Additionally, S4 exhibited a broad endothermic peak around 138 °C. This is due to the residual DMF solvent in the sample boiling and absorbing heat at this temperature.

From the DSC data of S1 and S2, it can be observed that neither S1 nor S2 exhibits significant exothermic behavior below 200 °C. This indicates that the exothermic behavior observed in S3, S4, and S5 in the range of 160~185 °C is not due to a single substance of AlNO or AlAC but rather results from the redox reaction between AlNO and AlAC: at this temperature, the nitrate ions in AlNO, along with a small amount of oxygen generated from the decomposition of nitrate ions, act as strong oxidants. They undergo vigorous redox reactions with AlAC, acting as a reducing agent, producing a large amount of heat, resulting in the formation of Al_2_O_3_ and a substantial quantity of gas. Significant mass loss observed in the TGA test results is attributable to this reaction. The heat provided by this rapid and intense exothermic reaction may promote the formation of M-O-M bonds and the removal of organic ligands, offering the potential for the preparation of high-performance dielectric layers at low temperatures.

In Figure 4, DSC data for other aluminum salts are provided for comparison. As seen in Figure 4a, a prominent endothermic peak appears around 130 °C, which is attributed to the removal of H_2_O in sample Al(NO_3_)_3_·8H_2_O [30]. In contrast, no similar endothermic peak appears around 130 °C in Figure 3a, as the DSC samples in this article were first dissolved in DMF and then dried to form solid samples, so the H_2_O was already dissolved in the DMF. However, none of the aluminum salts in Figure 4 exhibit significant exothermic peaks, which further indicates that the prominent exothermic peaks observed in Figure 3c–e are not caused by a single aluminum salt. Instead, they are due to the redox reactions between AlNO and AlAC, which release a substantial amount of heat.

### 3.3. Surface Morphology Analysis

As shown in Figure 5, the surface morphology of different film samples was observed using LSCM. From Figure 5a,c,d, the film surfaces are smooth and uniform, with no visible cracks or black particles. In contrast, Figure 5b,e clearly shows a significant distribution of black particles on the film surfaces, leading to an uneven texture. This may be due to the low solubility of AlAC in the DMF solvent: as the temperature increases, AlAC precipitates out before it has a chance to react, resulting in the formation of crystalline particles on the film surface.

Surface morphology of the thin films was scanned using AFM, as show in Figure 6, and surface roughness was calculated, as shown in Figure 7. The roughness of all five samples of dielectric films is less than 1 nm, indicating good uniformity of the dielectric films prepared by the solution method, providing potential for the fabrication of high-performance devices.

From Figure 6, it can be observed that both samples of non-self-exothermic system dielectric film surfaces exhibit relatively high roughness and numerous surface pores. This is due to the insufficient annealing temperature, resulting in a lower degree of film densification and, thus, the formation of more pores. In contrast, the surface roughness of the dielectric films in the three self-exothermic system samples is lower than that of the non-self-exothermic system samples. This indicates that the significant heat generated during annealing in the self-exothermic system contributes to an increased level of film densification. The surface roughness of samples S3 and S5 reached 0.747 ± 0.045 nm and 0.799 ± 0.056 nm, respectively. Meanwhile, the surface roughness of sample of S4 reached the lowest at 0.590 ± 0.017 nm, demonstrating the best performance. From Figure 6e, it can also be observed that there is a significant amount of particle precipitation on the surface of the S5 dielectric film. This is consistent with the results of LSCM testing, indicating that a higher content of AlAC leads to easier precipitation during annealing.

### 3.4. Thin Film Thickness Measurement

The insulation performance is closely related to the thickness, roughness, and density of the dielectric films [33]. XRR was used to fit the thickness and density, as shown in Figure 8. It is found that the thickness of samples of S1, S2, S3, S4 and S5 is 8.772 nm, 9.322 nm, 10.328 nm, 9.325 nm, and 7.872 nm.

### 3.5. Transmittance Test

The transmittance of the dielectric films was measured using UV-vis, and the test results are shown in Figure 9. Within the visible light range (400–760 nm), all five samples of films exhibited transmittance rates of over 98%, demonstrating excellent optical performance.

### 3.6. Leakage Current Density Analysis

Figure 10 displays the I–V testing curves of five sets of dielectric thin films. Within the tested range, the leakage current density of S3 and S4 is lower than that of S1 and S2, with S3 exhibiting the lowest leakage current density in the 0 to 5 MV/cm range. This indicates that a self-exothermic reaction contributes to the formation of M-O-M bonds within the film and the reduction in defect state density. Conversely, at higher electric fields, the leakage current density of S5 approaches that of S1 and S2, which may be attributed to the higher surface roughness of the S5 film, leading to increased defects in the contact area between the electrode and the film, thereby increasing the leakage current. Furthermore, the zero points of the IV curves for S3 are primarily located around 0 MV/cm, whereas the zero points for S1, S2, and S4 are shifted to around −2 to −1 MV/cm. This also indicates that a self-exothermic reaction helps to reduce internal defects within the film, thereby suppressing zero-point drift.

When an electric field of 1 MV/cm is applied, the leakage current densities of the three exothermic system (S3, S4, and S5) devices are 1.06 × 10^−8^ A/cm^2^, 3.81 × 10^−8^ A/cm^2^, and 1.86 × 10^−8^ A/cm^2^, respectively.

### 3.7. Capacitance Density Analysis

Capacitance density (C_i_) is one of the important parameters of the dielectric layer, reflecting the capacity of storing electrical charge per unit area of the film. The results of the *C*-V tests are shown in Figure 11. From the results, within the test range, S3 and S4 exhibit good capacitance–voltage stability and higher C_i_. However, the C_i_ of the two non-exothermic system devices is smaller, indicating that the significant heat released by the exothermic process benefits the formation of M-O-M bonds within the dielectric film. During the experiment, the C_i_ of S5 was so low that it exceeded the instrument’s range, resulting in a value close to zero. Combined with the unstable C_i_ observed in S2 and the results of surface roughness tests, we can conclude that even with exothermic reactions occurring, the excessive surface roughness of the film significantly affects the performance and stability of the devices.

The relative dielectric constant of dielectric films is further investigated. *C*-F tests were conducted on MIM devices. The relative dielectric constant can be calculated based on the *C*-F test results and the thickness of dielectric films, which can be calculated as follows:(2)$$C=\varepsilon_{0}k\frac{S}{d}$$
where ε_0_ is the vacuum permittivity (8.85 × 10^−12^ F/m), *d* represents the thickness of films, and s denotes the area of Al electrode (1.256 × 10^−3^ cm^2^). The relative dielectric constants @ 1 kHz of dielectric films are summarized in Figure 12. The relative dielectric constants of the three exothermic system films are all greater than those of the other two non-exothermic system films, which once again demonstrates the promoting effect of exothermic reactions. S3 exhibits the highest relative dielectric constant (8.61 ± 0.06) and the smallest standard deviation, indicating excellent dielectric performance and uniformity of the film.

Finally, I have listed a comparison of the key performance characteristics between the films discussed in this article and those produced using the most advanced methods currently available in Table 2. From the table, the Al_2_O_3_ films in this study exhibit superior performance in terms of leakage current density and dielectric constant compared to those prepared using auxiliary equipment. At the same time, the Al_2_O_3_ films in this article show performance in leakage current density that is comparable to films prepared via vacuum deposition methods and even surpasses the performance of some individual films. Therefore, the low-temperature self-exothermic reaction method for preparing dielectric films based on solution processing presented in this article is not only easy to operate and requires simple equipment but also demonstrates excellent performance, showing great potential. However, this method still faces several issues and challenges. Firstly, since the method is based on solution processing, the evaporation of the solvent during the annealing process can result in the formation of numerous pores of varying sizes within the film. This may lead to reduced film density and adhesion to substrates and increase leakage current density. Secondly, the organic solvents used in this study cause the formation of more pores on the film’s surface during annealing, which increases the surface roughness and thus affects the electrical properties of the film.

## 4. Conclusions

This paper employs a low-temperature self-exothermic reaction based on the solution method to fabricate high-performance transparent Al_2_O_3_ dielectric films at an annealing temperature of 185 °C. The experiments demonstrate that mixed systems of AlNO and AlAC in varying proportions undergo vigorous redox reactions and release significant heat at around 160~180 °C. During the annealing process, a higher content of AlAC results in more pronounced surface precipitation and higher surface roughness of the film. Therefore, based on comprehensive analysis, S3 exhibits the best performance: the AFM test results indicate a surface roughness of 0.747 ± 0.045 nm; the transmittance within the visible light range reaches up to 98%; the device leakage current density is as low as 1.06 × 10^−8^ A/cm^2^@1 MV/cm, with a high relative dielectric constant of 8.61 ± 0.06, demonstrating excellent electrical characteristics and stability. In summary, the alumina films prepared using the exothermic solution method reported in this paper exhibit lower annealing temperatures and surface roughness, higher transparency, lower leakage current density, and higher relative dielectric constant. These characteristics offer great promise for compatibility with flexible substrates and the fabrication of high-performance transparent electronic devices.

## Figures and Tables

**Figure 1 micromachines-15-01140-f001:**
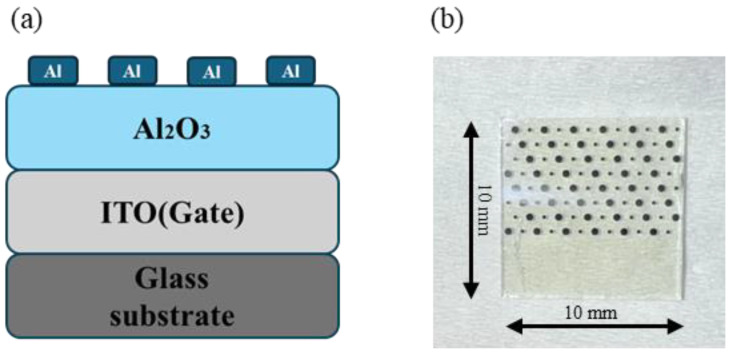
MIM device: (**a**) schematic; (**b**) photo.

**Figure 2 micromachines-15-01140-f002:**
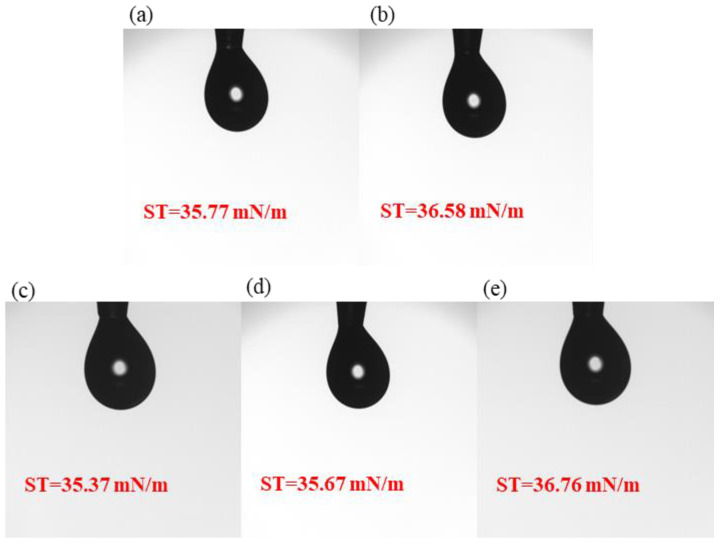
Surface tension: (**a**) S1; (**b**) S2; (**c**) S3; (**d**) S4; (**e**) S5.

**Figure 3 micromachines-15-01140-f003:**
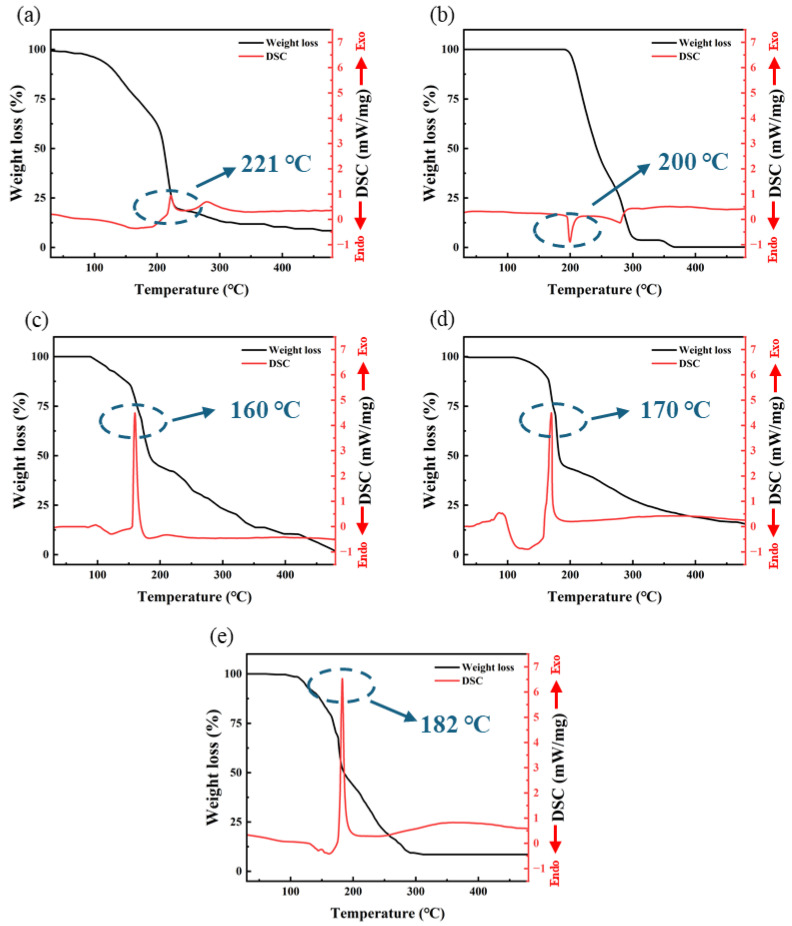
The TG-DSC test curve: (**a**) S1; (**b**) S2; (**c**) S3; (**d**) S4; (**e**) S5.

**Figure 4 micromachines-15-01140-f004:**
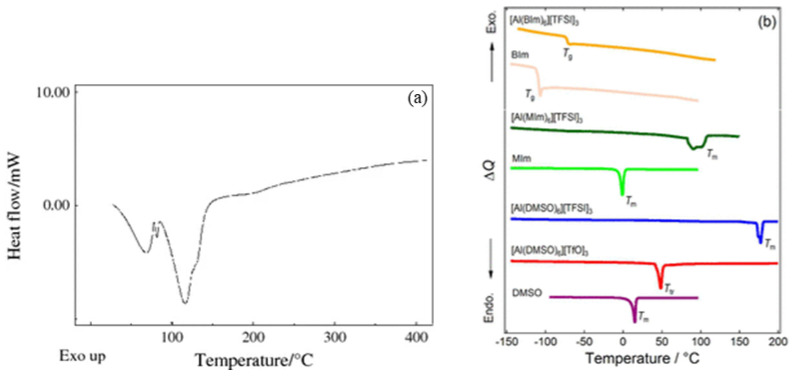
Comparison of DSC curves for different aluminum salts: (**a**) DSC curve of Al(NO_3_)_3_·8H_2_O [31]; (**b**) DSC traces of [Al(L)_6_]X_3_ (L = DMSO, MIm, or BIm; X = TFSI or TfO) with the pure ligands as references. T_m_, T_tr_, and T_g_ denote melting points, solid-solid phase transition temperatures, and glass transition temperatures, respectively [32].

**Figure 5 micromachines-15-01140-f005:**
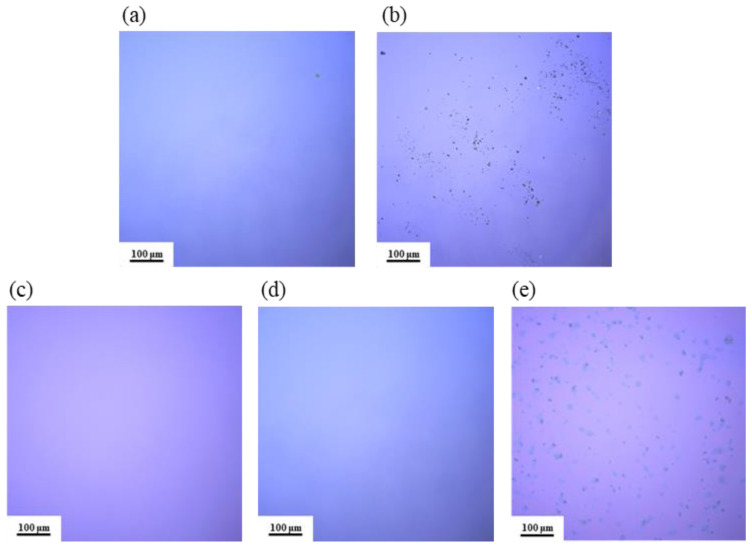
LSCM images of Al_2_O_3_ films: (**a**) S1; (**b**) S2; (**c**) S3; (**d**) S4; (**e**) S5.

**Figure 6 micromachines-15-01140-f006:**
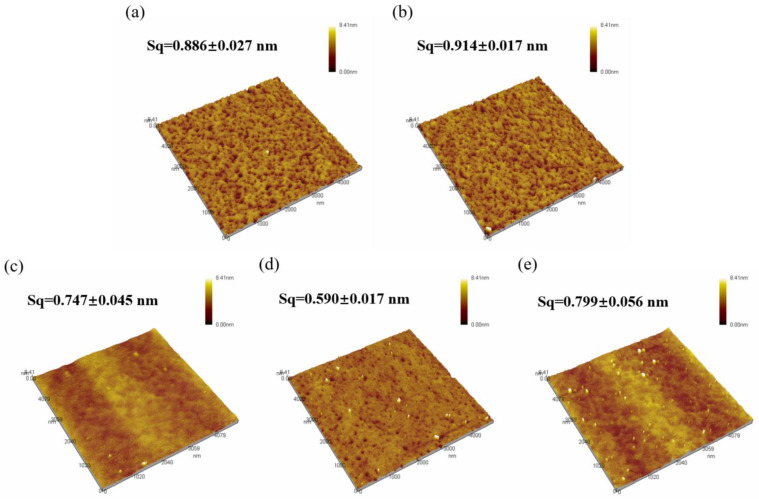
AFM morphology of Al_2_O_3_ films: (**a**) S1; (**b**) S2; (**c**) S3; (**d**) S4; (**e**) S5.

**Figure 7 micromachines-15-01140-f007:**
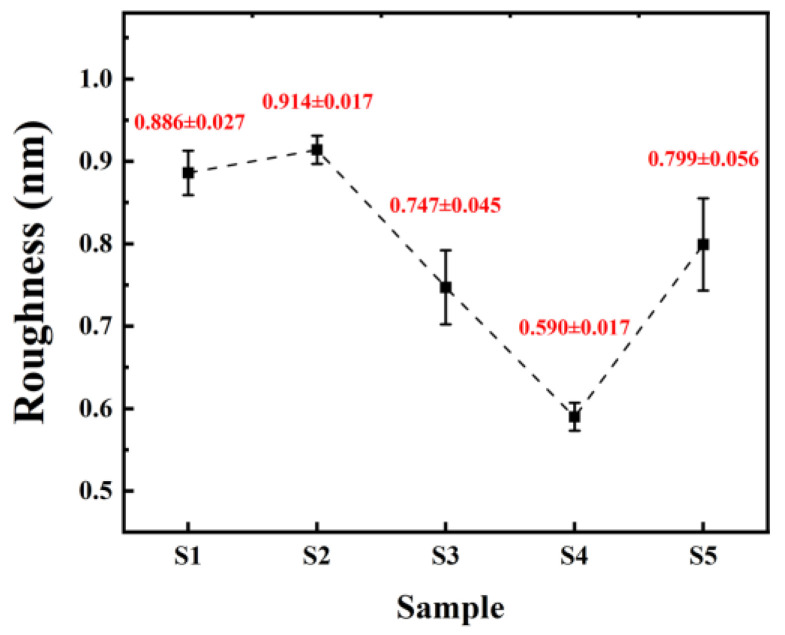
Surface roughness of different samples.

**Figure 8 micromachines-15-01140-f008:**
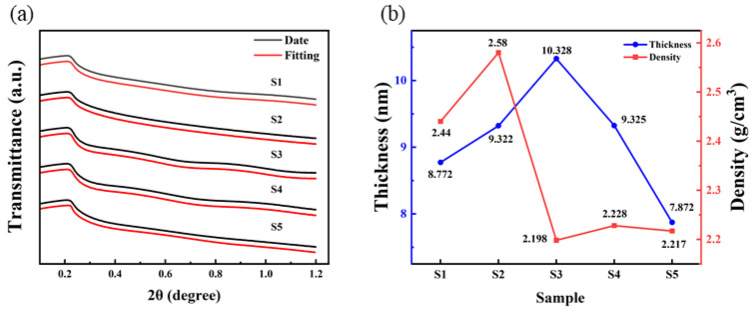
Al_2_O_3_ films: (**a**) XRR date and fitting curves; (**b**) thickness and density fitting results.

**Figure 9 micromachines-15-01140-f009:**
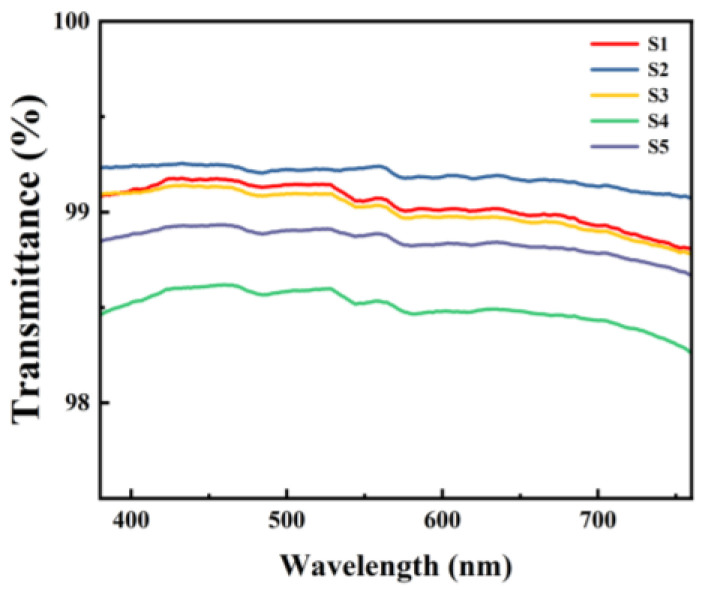
UV-Vis spectrum of different Al_2_O_3_ films.

**Figure 10 micromachines-15-01140-f010:**
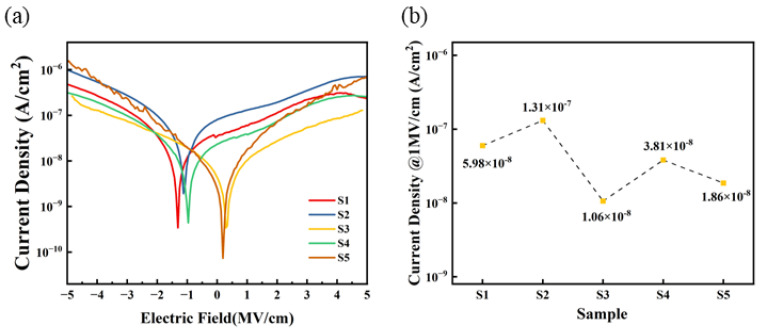
Al_2_O_3_ dielectric films: (**a**) leakage current density curves; (**b**) summary of leakage current density @1 MV/cm.

**Figure 11 micromachines-15-01140-f011:**
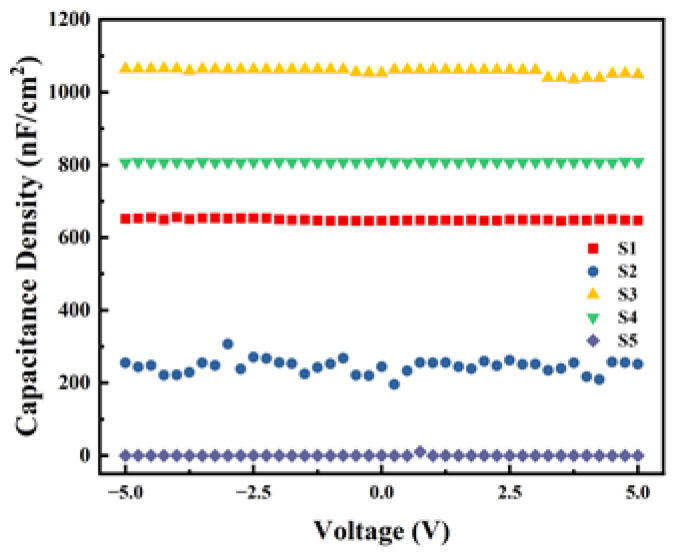
Capacitance density test curves of different samples.

**Figure 12 micromachines-15-01140-f012:**
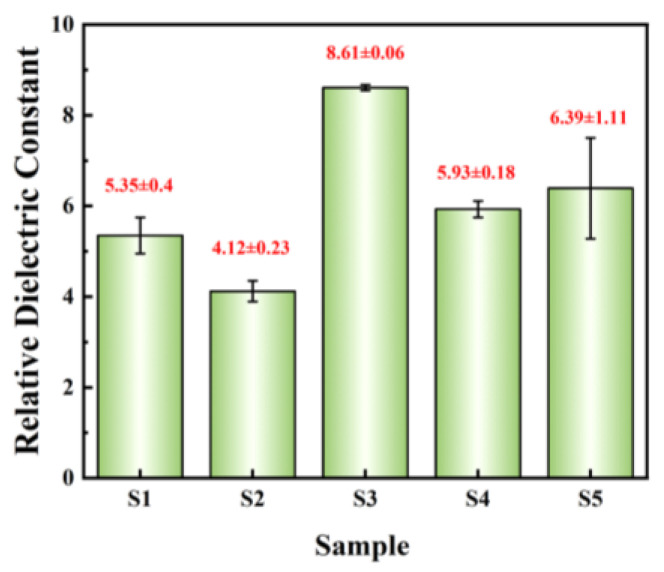
Summary of relative dielectric constant @1 kHz.

**Table 1 micromachines-15-01140-t001:** Proportion of precursor system components and naming of precursor systems.

Proportion of Precursor System Components	c(AlNO)/mol	c(AlAC)/mol	Naming
Pure AlNO	0.2000	0	S1
Pure AlAC	0	0.2000	S2
AlNO:AlAC = 2:1	0.1333	0.0667	S3
AlNO:AlAC = 1:1	0.1000	0.1000	S4
AlNO:AlAC = 1:2	0.0667	0.1333	S5

**Table 2 micromachines-15-01140-t002:** A thorough quantitative comparison of leakage current density and dielectric constant against state-of-the-art methods.

Dielectric Material	Preparation Method	Leakage Current Density (A/cm^2^)	Relative Dielectric Constant	Refs.
Al_2_O_3_	self-exothermic reaction	1.06 × 10^−8^ @1 MV/cm	8.61 @1000 Hz	this article
Al_2_O_3_	DUV assisted exothermic process	2.9 × 10^−7^ @1 MV/cm	8.6 @1000 Hz	[23]
Al_2_O_3_	aqueous route	4.68 × 10^−9^ @1 MV/cm	8.6 @100 Hz	[24]
Al_2_O_3_	infrared irradiation	3.5 × 10^−8^ @1 MV/cm	7.6 @1000 Hz	[25]
Al_2_O_3_/TiO_2_/Al_2_O_3_ (6 nm/40 nm/6 nm)	PLD	2.03 × 10^−8^ @3 V		[34]
Al_2_O_3_/HfO_2_/Al_2_O_3_	ALD	10^−8^~10^−9^ @1 MV/cm	20.7	[35]
Al_2_O_3_	PEALD	10^−9^ @1 MV/cm	9.3 @100 kHz	[36]

## Data Availability

The data are contained within the article.

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
