# Peer review of "Preparation of High-Performance Transparent Al2O3 Dielectric Films via Self-Exothermic Reaction Based on Solution Method and Applications"

_micromachines, 2024, doi:10.3390/mi15091140_

Round 1

Reviewer 1 Report

Comments and Suggestions for Authors

My main comments are

1. I consider the keywords should more clearly reflect the content of the research. They are keywords not key sentences. Moreover, “Low-temperature self-exothermic solution”. Solution can not be exothermic.

Al(NO3)3 and Al(Acac)3 are, I believe, precursors, not keywords.

2. “Exothermic synthesis reaction, exothermic synthesis process and others ” It is better “exothermic reaction, exothermic process”

3. What does “self-exothermic” mean? It is due to exclusively the fact that it “does not require additional equipment assistance”, isn’t it? Why do dear authors introduce this neologism?

4. The respected authors should more clearly formulate the novelty and purpose of their research. As I can see, the novelty and goal lie exclusively in the use of the chemical reaction heat. And what about the prize of Al(Acac)3?

5. As for Table 1, I consider this Table is not necessary. The names of the mixtures are also very strange. They resemble quasi-chemical formulas with subscripts. I think that it is better to use more simple names, e.g., L1, L2, L3 or S1 (Solution1), S2, S3 and so on. Along with, they could be used directly when they first appear in the manuscript body.

6. Lines 94-95. What are the ratios used? What are 0, 1, 2 and so on? How these values were calculated? Without this information, results will not be reliable. These data are more important than what sort of filter was used (line 98).

7. Empirical formulas of aluminum acethylacetonate (C15H21O6Al) and N,N-dimethylethanolamine (C3H7NO) are wrong (line 91). The respected authors are encouraged to correct.

8. “The total concentration of aluminum ions in each sample is controlled at 0.2 M” What does it mean?

9. Abbreviations are deciphered once when they first appear.

10. English requires to improve (including grammar errors). There are phrases poor understandable, e.g., lines 147-149.

11. One has to show a direction of exo-process with arrows on DSC curves.

12. Why are exothermic signals on DSC curves attributed to decomposition?

13. The DSC data on aluminum salts are very well known. The references are needed for comparison.

14. Low quality of Fig. 4. There are seen no features.

15. The data should be correctly presented, e.g., instead of “roughness 0.747±0.04 nm» it should be “roughness 0.75±0.04 nm” and others.

16. Why does Fig. 10 show the electrochemical data for only one sample? It is interesting to compare the data for all samples.

Comments on the Quality of English Language

It seems to me, the English language requires improvements 

Reviewer 2 Report

Comments and Suggestions for Authors

This manuscript presents  a self-exothermic solution method for fabricating high-performance aluminum oxide dielectric films, which addresses a critical challenge in integrated circuit design: reducing production cost and preparation cycle while maintaining high dielectric performance. The approach’s compatibility with flexible substrates, low annealing temperature, and high transparency offers significant advantages for next-generation electronic devices. I am favorable towards the publication of this manuscript but recommend to address the following issues to solidify its contribution:

1. Lack of quantitative comparison: The paper does not provide a thorough quantitative comparison of the key performance metrics (e.g., leakage current density, dielectric constant) against state-of-the-art methods such as vacuum deposition or other solution-based approaches (e.g., UV-assisted or infrared-assisted methods).

2. No discussion on process trade-offs: The authors do not discuss trade-offs such as whether the simpler self-exothermic method compromises other important properties like long-term stability, adhesion to substrates, or mechanical durability compared to more established methods.
